# Numerical Analysis of a CZTS Solar Cell with MoS_2_ as a Buffer Layer and Graphene as a Transparent Conducting Oxide Layer for Enhanced Cell Performance

**DOI:** 10.3390/mi13081249

**Published:** 2022-08-03

**Authors:** Sampad Ghosh, Samira Yasmin, Jannatul Ferdous, Bidyut Baran Saha

**Affiliations:** 1Department of Electrical and Electronic Engineering, Chittagong University of Engineering and Technology (CUET), Chattogram 4349, Bangladesh; sampad@cuet.ac.bd (S.G.); u1602029@student.cuet.ac.bd (S.Y.); u1602020@student.cuet.ac.bd (J.F.); 2Department of Mechanical Engineering, Kyushu University, 744 Motooka, Nishi-ku, Fukuoka 819-0395, Japan; 3International Institute for Carbon-Neutral Energy Research (WPI-I2CNER), Kyushu University, 744 Motooka, Nishi-ku, Fukuoka 819-0395, Japan

**Keywords:** CZTS solar cell, graphene, molybdenum disulfide (MoS_2_), SCAPS simulation

## Abstract

Copper zinc tin sulfide (CZTS) can be considered an important absorber layer material for utilization in thin film solar cell devices because of its non-toxic, earth abundance, and cost-effective properties. In this study, the effect of molybdenum disulfide (MoS_2_) as a buffer layer on the different parameters of CZTS-based solar cell devices was explored to design a highly efficient solar cell. While graphene is considered a transparent conducting oxide (TCO) layer for the superior quantum efficiency of CZTS thin film solar cells, MoS_2_ acts as a hole transport layer to offer electron–hole pair separation and an electron blocking layer to prevent recombination at the graphene/CZTS interface. This study proposed and analyzed a competent and economic CZTS solar cell structure (graphene/MoS_2_/CZTS/Ni) with MoS_2_ and graphene as the buffer and TCO layers, respectively, using the Solar Cell Capacitance Simulator (SCAPS)-1D. The proposed structure exhibited the following enhanced solar cell performance parameters: open-circuit voltage—0.8521 V, short-circuit current—25.3 mA cm^−2^, fill factor—84.76%, and efficiency—18.27%.

## 1. Introduction

At present, the world’s energy demand is greatly dependent on fossil fuels. However, this limited stock of fossil fuels will run out very soon. Moreover, it is not environmentally friendly because it emits greenhouse gases. Hence, if alternative or renewable sources are not explored, the world will face a severe energy crisis in the upcoming future. Renewable energies are clean sources of energy, as they emit no harmful elements. Among the renewable energy sources, solar is the fastest growing because of its easy adoption. Solar cells, commonly noted as PV cells or photovoltaic cells, transform sun-powered energy directly into electricity. There are many forms of solar cells. Silicon (Si) solar cells are today’s most widely used cells, as they are available at a reasonable price and can offer a good efficiency of 26.7% against an intrinsic limit of 29% [1,2].

Thin-film solar cells are another type of photovoltaic cell where a very thin layer of semiconductor materials, for instance, copper indium gallium diselenide (CIGS) and cadmium telluride (CdTe), is used. They are flexible and lightweight due to their attractive thickness. Hence, they are more economical than Si solar cells [3]. Though group III–V solar cells show a more attractive efficiency, they have higher manufacturing costs. Moreover, research has been conducted on organic, quantum dots, and perovskite solar cells to offer more efficient solar cells. Thin film solar cells have gained popularity due to their low material consumption. Though CdTe- and CIGS-based solar cells are industrially successful [4], there are some restrictions on using them. Indium in CIGS is a rare material [5] and tellurium in CdTe is not only rare but also mildly toxic [6]. This is why looking for a cost-effective, environment-friendly, naturally abundant material that provides an adequate conversion efficiency absorber layer is of great interest. Ultra-thin copper zinc tin sulfide (Cu_2_ZnSnS_4_), also known as CZTS, can be a good choice that may fulfill all these requirements. Its components materials are non-toxic and earth-abundant. Furthermore, CZTS-based solar cells are highly efficient and reliable [7].

CZTS-based solar cell shows good optical and electrical properties with an absorption coefficient of 10^4^ cm^−1^, an electrical resistivity of around 10^−2^ ohm cm, and a band gap of 1.54 eV at room temperature [8,9]. According to the Shockley–Queisser limit, CZTS-based solar cells have a theoretical 32.4% conversion efficiency limit [10]. The highest power conversion efficiency of 12.2% was obtained recently for CZTS-based solar cells, which is significantly less than the theoretical limit. Therefore, there is still a huge scope to improve the efficiency and other parameters [11]. A numerical study showed that a solar cell with an Al:ZnO/i-ZnO/CdS/CZTS/Mo structure exhibits an efficiency of 15.84% [12], where Al-ZnO acts as a transparent conducting oxide (TCO) layer, CdS as a buffer layer, and Mo as the back contact. The intrinsic ZnO layer was grown between the buffer and TCO layer to ensure that the defective region of the absorber layer did not suppress the open-circuit voltage [13]. Conversely, ZnO/InSe/CZTS- and i-ZnO/MoS_2_/CZTS/Mo-based structures showed conversion efficiencies of 16.30% [7] and 17.03% [14], respectively, as obtained from simulation analyses. Furthermore, a maximum efficiency of 17.6% was obtained from FTO/ZnO/CdS/CZTS baseline solar cells [15]. Very recently, we numerically investigated and reported a solar cell efficiency of 17.14% for the graphene/ZnO/CZTS/Ni structure [16], which showed further enhancement of the conversion efficiency. In this CZTS solar cell structure, graphene, ZnO, and Ni are employed as a TCO layer, buffer layer, and back contact, respectively. In this current work, we introduced molybdenum disulfide (MoS_2_), considering their promising features as a buffer layer material for further investigation. 

The buffer layer’s role in solar PV is to reduce the defects and interfacial imperfections caused by the window layer and to make sure that the absorber layer and the window layer have the right band alignment. From the literature, it was found that cadmium sulfide (CdS) is usually used as the buffer layer material. However, this contains the toxic element cadmium (Cd) and it generates a significant amount of waste through the deposition process [17]. On the other hand, molybdenum disulfide (MoS_2_) belongs to the two-dimensional transition metal dichalcogenides (TMDCs) group, which exhibits excellent optical (absorption coefficient is 10^5^–10^6^ cm^−1^), mechanical, and electronic properties with a suitable bandgap of 1.3 eV. It has attractive optoelectronic properties, such as a tunable work function, an adjustable bandgap with reduced thickness, great mobility of 50 cm^2^ V^−1^ s^−1^, and potential interaction with light [18]. Moreover, graphene and MoS_2_ composites provide more active sites and improve the conductivity [19]. This feature also motivated us to incorporate MoS_2_ as a buffer layer with the TCO layer of graphene. Moreover, graphene is employed over conventional indium tin oxide (ITO) because they are more flexible, transparent, and economical. Furthermore, they provide attractive optical, mechanical, and thermal properties [20]. It has greater transparency of approximately 2.3% light absorption, a higher melting point of approximately 5000 K, and higher thermal conductivity of approximately 10^3^ Wm^−1^ K^−1^ [21]. In addition, at room temperature, suspended graphene has a higher carrier mobility of 2 × 10^5^ cm^2^ V^−1^ s^−1^.

In this research work, a highly efficient CZTS solar cell was designed by incorporating graphene (GnP) as a transparent conducting layer and MoS_2_ as a buffer layer. A systematic and thorough investigation was conducted using the solar cell simulator named SCAPS-1D. The influences of the buffer layer, thickness variation of CZTS, the doping concentration in CZTS, and thermal stability were analyzed and subsequently discussed in the following subsections for the proposed solar cell structure.

## 2. Materials and Methods

### 2.1. Structure of CZTS Solar Cell

Photons from solar radiation have different energies, where these energies can be turned into electricity by stacking different bandgap materials. The material’s bandgap should be big enough to absorb high-energy photons. Therefore, a layer with a bigger gap is used at the front. Usually, two types of solar cell structures are used: heterojunction and multi-junction. Figure 1 shows the proposed structure of a CZTS solar cell structure (GnP/MoS_2_/CZTS/Ni/glass substrate), which is based on a heterojunction solar cell.

### 2.2. Transparent Conducting Oxide or Window Layer

In optoelectronic and photovoltaic devices, a transparent conducting oxide (TCO) layer constructed of doped metal oxide is usually used. This layer is also known as the window layer. It has the lowest carrier concentration of 10^20^ cm^−3^, the highest conductivity of larger than 10^3^ S cm^−1^, and better than 80% transparency [22]. Given its desirable optical and electrical characteristics, high conductivity, and transparency (>90%), graphene was chosen as the window layer in this instance [23]. Moreover, we preferred graphene to the widely used window layer material indium tin oxide (ITO) because ITO needs expensive fabrication methods and it contains the rare element indium (In) [24]. It offers improved thermal stability compared with other traditional conducting oxide layers. Furthermore, the bandgap of graphene can be tuned with the addition of dopants such as boron or copper [25]. In our previous work, we optimized the thickness of graphene as a window layer at 2 nm, which ensured greater transparency [16]. Thus, the thickness of the window layer as 2 nm is deliberately used throughout the simulation of the proposed solar cell structure.

### 2.3. Buffer Layer

To form a p–n junction with the absorber layer, a buffer layer (single or double) is used. For the buffer layer, a more extensive bandgap layer is required to ensure maximal light transmission, little absorption loss, and minimal recombination loss or to convey the most photo-generated carriers possible to the outer circuit. Additionally, it has the ideal thickness to provide low series resistance. The open-circuit voltage (V_oc_) in solar cells is improved significantly by this buffer layer [26,27]. Here, molybdenum disulfide (MoS_2_) is selected as the buffer layer material. It was anticipated that the performance of solar cells can be increased by using an n-type semiconductor with a narrow direct bandgap of 1.3 eV compared with the graphene/ZnO/CZTS/Ni structure [16], where MoS_2_ will be used as a buffer layer instead of ZnO. Graphene/MoS_2_ composites have overcome the shortcomings of their respective counterparts owing to their beneficial physical or chemical properties. They constitute heterostructures in certain ways, where molybdenum disulfide (MoS_2_) and graphene have integral physical properties and possess parallel lattice structures. This mitigates the shortcomings of the respective counterparts and optimizes photovoltaic solar cell performance [19].

### 2.4. Absorber Layer

Another crucial and integral component of a solar cell that absorbs energy from natural or artificial light is the absorber layer. An effective absorber layer should absorb the radiation at wavelengths in the visible portion of the electromagnetic spectrum because the majority of the light energy is found here. Copper zinc tin sulfide (Cu_2_ZnSnS_4_) or CZTS thin film was taken into consideration as an absorber layer in the proposed structure because of its appealing absorption coefficient (>10^4^ cm^−1^) and good physical and electrical characteristics (bandgap of approximately 1.4 to 1.6 eV) [28]. They also solely offer an efficiency of more than 20% and are earth-friendly and abundant [25].

### 2.5. Back Contact

The back or rear contact plays a critical function in improving performance metrics and solar cell efficiency. Nickel (Ni) performs better than other materials when used as a back contact. The energy needed to remove electrons from the metal surface is known as the work function, which is 5.15 eV for nickel [29]. The performance of the device can be greatly enhanced by a stable ohmic contact, which can lower the back contact interface recombination. CZTS has a bandgap of 1.54 eV and an electron affinity of 4.5 [7]. As a result, a metal with a greater work function is required for a static ohmic contact, and Ni has just that.

### 2.6. Soda Lime Glass Substrate

Thin-film solar cells also need parts that are made of a substrate. The electron–hole recombination at the grain borders is prevented by the diffusion of Na from the soda-lime glass (SLG) substrate to the absorber layer’s grain boundaries. Because of this, alkali metal oxides (Na_2_O and K_2_O) were studied and made in a lab [30]. This SLG substrate, which is smooth and provides the thin-film solar cell mechanical support, is widely utilized for thin-film deposition. It is comparably inexpensive, chemically stable, and very useful in solar cell research.

### 2.7. Mathematical Modeling

In this work, simulation was carried out using SCAPS (version 3.3.0.9). SCAPS-1D (a Solar Cell Capacitance Simulator), developed by Burgleman et al. [31], is used to carry out simulations of solar cell structures; it is a one-dimensional solar cell simulator that enables the simulation of up to seven layers. The SCAPS-1D simulator helps to perform quick simulations and come up with batch calculations. It provides a user-friendly interface and helps to load and save all settings and data easily [32]. SCAPS is designed to simulate practical situations; hence, convergence failure and superficial output may occur when unrealistic parameters are input.

SCAPS usually performs simulations based on three groups of equations. The first one is transport equations for electrons and holes, which can be written as
(1)$$\frac{1}{q}\frac{dj_{n}}{dx}=G_{n}-U_{n}$$
(2)$$\frac{1}{q}\frac{dj_{p}}{dx}=G_{p}-U_{p}$$
where *J_n_*, *J_p_*, *G_n_*, *G_p_*, *U_n_*, and *U_p_* represent the electron current density, hole current density, the generation rate of electrons, the generation rate of holes, recombination rate of electrons, and recombination rate of holes, respectively. The second one is the Poisson equation, which is represented as
(3)$$\frac{d^{2}\emptyset }{dx^{2}}=\frac{q}{\in_{0}\in_{r}}\left(p\left(x\right)-n\left(x\right)+N_{d}-N_{a}+\rho_{p}-\rho_{n}\right)$$
where *ϕ*(*x*), *p*(*x*), *n*(*x*), Ԑ_o_, Ԑ*_r_*, *q*, *N_d_*, *N_a_*, *𝜌_𝑛_*, 𝑎𝑛𝑑 *𝜌_𝑝_* are the electrostatic potential, hole concentration, electron concentration, vacuum permittivity, relative permittivity, electric charge, charge impurities of donor, charge impurities of the acceptor, electron distribution, and hole distribution, respectively. Finally, the third group of equations contains the drift and diffusion equations, which can be formulated as
(4)$$j_{n}=D_{n}\frac{dn}{dx}+\mu_{n}n\frac{d\varphi }{dx}$$
(5)$$j_{p}=D_{p}\frac{dp}{dx}+\mu_{p}p\frac{d\varphi }{dx}$$
where *D_n_*, *μ_n_*, *D_p_*, and *μ_p_* are the diffusion coefficient of electrons, mobility of electrons, the diffusion coefficient of holes, and mobility of holes, respectively [32,33]. All these equations are used to calculate solar cell performance parameters via the SCAPS simulator.

### 2.8. Numerical Simulation and Device Modeling

Numerical modeling and simulation are required before the production process to ensure the performance and stability of the proposed cell. The simulation settings of the layers determine the cell’s performance. Different characteristics of a layer, such as the thicknesses of buffer and absorber layers and the doping concentration of the absorber layer were varied accordingly to study the cell’s performance. The influence of temperature was also investigated to assess the cell’s endurance and thermal stability. Three layers (TCO, buffer, and absorber) of the proposed structure were modeled using the SCAPS 3309 tools. Necessary electrical and optical parameters of different layer materials were obtained from the literature [13,15,21,23,34,35,36,37,38,39,40] for their reasonable estimation during the simulation, as described in Table 1.

## 3. Results and Discussion

### 3.1. Effect of Buffer Layer Thickness

With a bandgap of about 1.3 eV, MoS_2_ was used as a buffer layer to ensure that the majority of incident light was directed toward the junction. The n-layer thickness should ideally be as thin as possible to improve the device’s series resistance. A reduced buffer layer thickness results in better short-circuit current density and minimal absorption in the blue region of the AM1.5 sun spectrum.

Simulations were carried out by varying the MoS_2_ layer thickness from 0.02 µm to 0.18 µm, and the effects on performance metrics were noted. The simulated results are shown in Figure 2. It was evident from the simulation findings that the thickness of MoS_2_ had a considerable impact on both the short-circuit current density (J_sc_) and efficiency (η). In this context, the optimized thickness of MoS_2_ was chosen to be 0.04 µm since above this thickness, the efficiency did not increase significantly, and a higher thickness increased the number of ionizing photons, resulting in more carriers. Moreover, the J–V curve for the variation of buffer layer thickness is shown in Figure 3. It was found that both the short-circuit current density and open-circuit voltage increased with the buffer layer thickness. Hence, the efficiency increased. This was because the MoS_2_ layer ensured a good p–n junction with the p-type CZTS absorber layer.

### 3.2. Effect of Absorber Layer Thickness

One of the major goals of this research was to improve cell performance by preserving the material and optimizing the CZTS absorber layer thickness. Keeping this in mind, we simulated the CZTS solar cell structure and the results are presented in Figure 4, where the absorber layer thickness varied from 0.5 µm to 4 µm. Figure 4 also presents the effect of absorber layer thickness on the key performance indicators of a solar cell. These include the open-circuit voltage (V_oc_) in V, short-circuit current density (J_sc_) in mA cm^−2^, fill factor (FF) in percent, and power conversion efficiency (η) in percent.

As shown in Figure 4, the efficiency (η) of the designed cell increased significantly as the CZTS absorber layer’s thickness increases up to 2 µm. After that, increasing the thickness of the absorber layer did not effectively boost the efficiency. However, the other parameters, such as V_oc_, J_sc,_ and FF had an increasing trend with the increase in the absorber layer thickness. Hence, 2 µm was considered to be the optimized absorber layer thickness that can contribute to obtaining optimal efficiency. Moreover, considering a thinner absorber layer could reduce the fabrication cost of CZTS solar cells. We also added the J–V characteristic curve for the variation in absorber layer thickness, as shown in Figure 5. This figure indicates that the short-circuit current density and open-circuit voltage had a significant effect on the increase in absorber layer thickness. It was hypothesized that a thicker absorber layer allows more photons to enter, resulting in more electron–hole pair generation.

### 3.3. Effect of Doping Density of CZTS Absorber Layer

Using the SCAPS-1D simulation software, many trials were conducted to determine how different doping concentrations of the CZTS absorber layer in solar cells could be used. The doping density was varied between 1 × 10^11^ cm^−3^ and 1 × 10^18^ cm^−3^ to investigate their effect on the solar cell parameters, as shown in Figure 6. From Figure 6, it is found that with the increase in CZTS doping concentration, except for the short-circuit current density (J_sc_), all other parameters (V_oc_, FF, and η) increase, indicating their dependence on the doping density of the CZTS absorber layer. The decrease in J_sc_ was due to the increase in the recombination of photogenerated carriers. Alternatively, the relationship between the open-circuit voltage (V_oc_) and short-circuit current density (J_sc_) with doping density could be explained by the following equations.

Considering a solar cell with a p–n junction diode, the well-known diode equation can be written as
(6)$$I=I_{0}\left(e^{\frac{qV}{kT}}-1\right)-I_{L}$$
where *I*, *I*_0_, *I_L_*, *q*, *V*, *k*, and *T* denote the net current flowing through the junction, the diode leakage current density in the absence of light, load current, electron charge, the voltage across the p–n junction, Boltzmann’s constant, and temperature, respectively. Furthermore, *I*_0_ can be represented by
(7)$$I_{0}=qA\left(\frac{D_{n}}{L_{n}}\frac{n_{i}^{2}}{N_{A}}+\frac{D_{p}}{L_{p}}\frac{n_{i}^{2}}{N_{D}}\right)$$
where *A*, *D_n_*, *D_p_*, *L_n_*, *L_p_*, *N_A_*, *N_D_*_,_ and *n_i_* signify the cross-sectional area of the p–n junction, the diffusion coefficient of the electron, the diffusion coefficient of the hole, the diffusion length for an electron, the diffusion length for a hole, the concentration of acceptor atoms, the concentration of donor atoms, and intrinsic carrier concentration, respectively. In contrast, the mathematical equation for the open-circuit voltage (*V_oc_*) is given as
(8)$$V_{oc}=\frac{kT}{q}\ln\left(\frac{I_{L}}{I_{o}}+1\right)$$

Putting *V* = 0 into Equation (6) provides the short-circuit current (I_sc_) from which the short-circuit current density (J_sc_) can be evaluated. It is seen from the above equations that *V_oc_* and J_sc_ are strongly dependent on the carrier doping density. As shown in Figure 6, above a carrier density of 0.1 × 10^17^ cm^−3^, the solar cell parameters did not improve significantly. Hence, the optimized value of doping density was 0.1 × 10^17^ cm^−3^ for the proposed cell structure. Figure 7 shows the J–V curve that was generated by varying the doping density of the absorber layer. From this figure, it can be seen that increasing the doping concentration decreased the short-circuit current density. Since the open-circuit voltage increased with the doping concentration, there was an improvement in the overall efficiency.

### 3.4. Effect of Temperature

The efficacy of solar cells is typically negatively impacted by increased temperatures [35]. Here, the working temperature was considered to be 300 K (27 °C). Solar cells are used outside, where temperature fluctuations might affect the output. Excessive heat may degrade the performance as well. The thermal stability of the CZTS-based solar cell was explored within the temperature range of 290 K to 380 K. This helped to examine the performance of the designed solar cell at various operating temperatures, as shown in Figure 8.

The drop in the efficiency (η) of the cell was due to the decrease in the open-circuit voltage (V_oc_) with increasing cell temperature. It was observed that current density (J_sc_) was almost unchanged throughout the temperature range, indicating no impact with temperature variation. A solar cell’s thermal stability is indicated by the temperature coefficient, which also demonstrates how the solar cell output varies with temperature. From Figure 8, it can be illustrated that there was a declining marginal trend for other output parameters (V_oc_, FF, and η). However, these were not greatly altered. Therefore, it can be concluded that the designed structure exhibited improved thermal stability.

### 3.5. Quantum Efficiency (QE)

Quantum efficiency can be known as the proportion of collected charge carriers to photons incident onto a solar cell. Due to the generation of an electron–hole pair from each photon, the quantum efficiency could theoretically be 100%. However, this does not occur in actual cells owing to many types of losses, such as buffer and window layer absorption, absorber layer absorption restriction, deep penetration, and recombination loss [35,40]. The quantum efficiency of the simulated solar cell structure is depicted in Figure 9. It demonstrates that the designed structure achieved a quantum efficiency of close to 90%. Therefore, it can be stated that the designed solar cell structure could maximize the utilization of solar irradiance.

### 3.6. Current Density–Voltage (J–V) Characteristics

The designed structure of graphene/MoS_2_/CZTS/Ni solar cell exhibits enhanced current density–voltage (J–V) characteristics, as revealed in Figure 10. In Figure 10, the proposed structure shows superior J–V properties and performance. The reason behind the better performance of the solar cell with MoS_2_ as the buffer material is its higher absorption coefficient and, consequently, more photogenerated carriers. Table 2 provides a summary of the optimal results attained in this investigation. This table also includes other solar cell performance parameters that were obtained from the numerical study that are available in the literature for further comparison. This table shows that our obtained results were comparable to the literature’s experimental values. From this, we can infer that the SCAPS-1D simulator is a capable tool for forecasting solar cell behavior in actual scenarios.

## 4. Conclusions

Using the SCAPS-1D program, we numerically analyzed CZTS solar cells by considering MoS_2_ as the buffer layer and graphene as the transparent conducting oxide (TCO) or window layer. Our objective was to select MoS_2_ as the buffer material for the CZTS absorber layer. The simulation results revealed that MoS_2_ as a buffer layer was suitable for the CZTS absorber layer. We then investigated the influence of the absorber layer’s thickness and doping density on the selected heterojunction structure. We were able to obtain the best open-circuit voltage (V_oc_), short-circuit current density (J_sc_), fill factor (FF), and efficiency (η) by optimizing these two factors. The optimized values for the buffer layer thickness, absorber layer thickness, and doping density were 40 nm, 2 µm, and 0.1 × 10^17^ cm^−3^, respectively. These adjusted settings allowed us to increase the efficiency of each heterojunction significantly. Indeed, we obtained a significantly higher efficiency of 18.27% (V_oc_ = 0.8521 V, J_sc_ = 25.3 mA cm^−2^, and FF = 84.76%) for the proposed CZTS solar cell structure. The obtained results were satisfactory and may be employed experimentally to fabricate actual graphene/MoS_2_/CZTS-based solar cells in the future.

## Figures and Tables

**Figure 1 micromachines-13-01249-f001:**
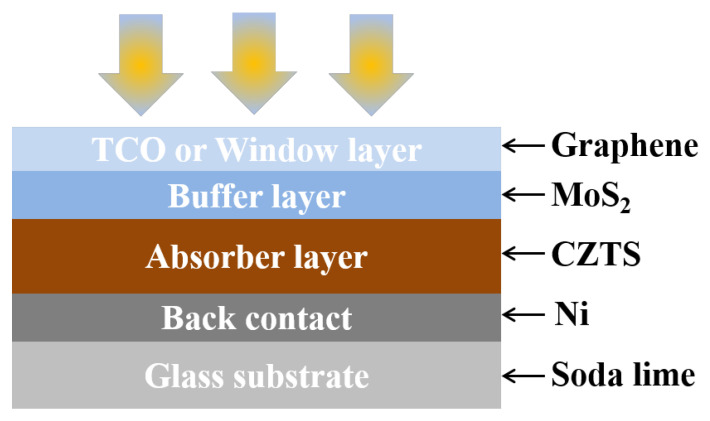
The CZTS-based solar cell structure.

**Figure 2 micromachines-13-01249-f002:**
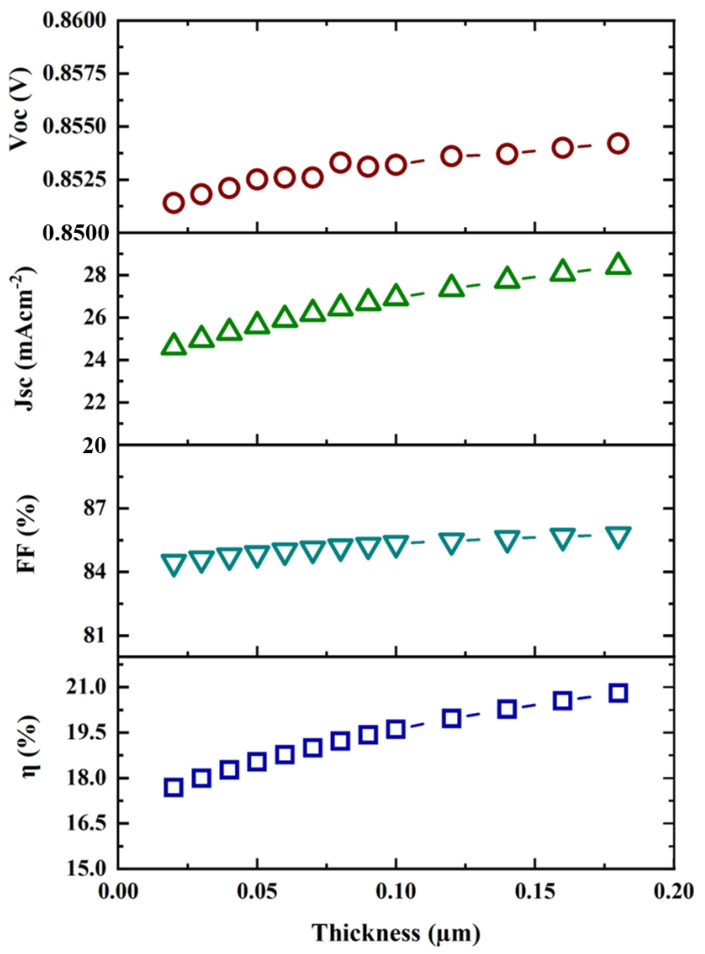
Buffer layer thickness versus V_oc_, J_sc_, FF, and η.

**Figure 3 micromachines-13-01249-f003:**
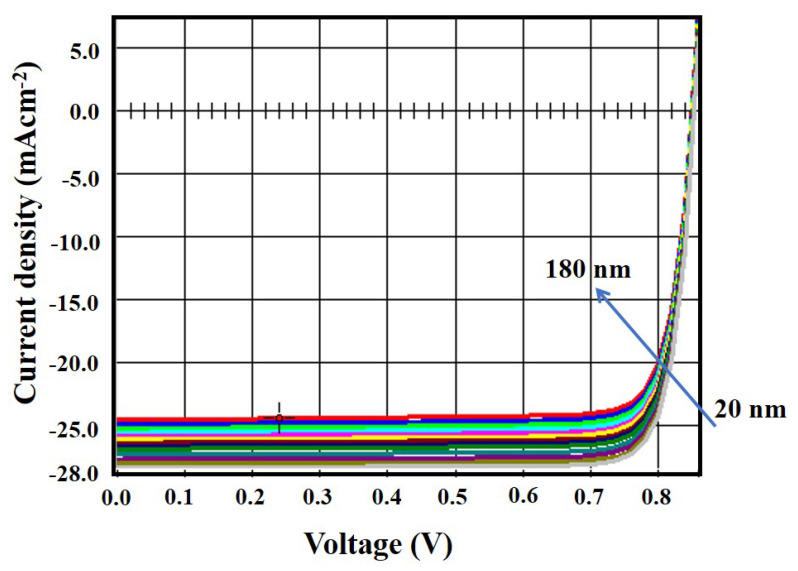
J–V curve with the variation of buffer layer thickness.

**Figure 4 micromachines-13-01249-f004:**
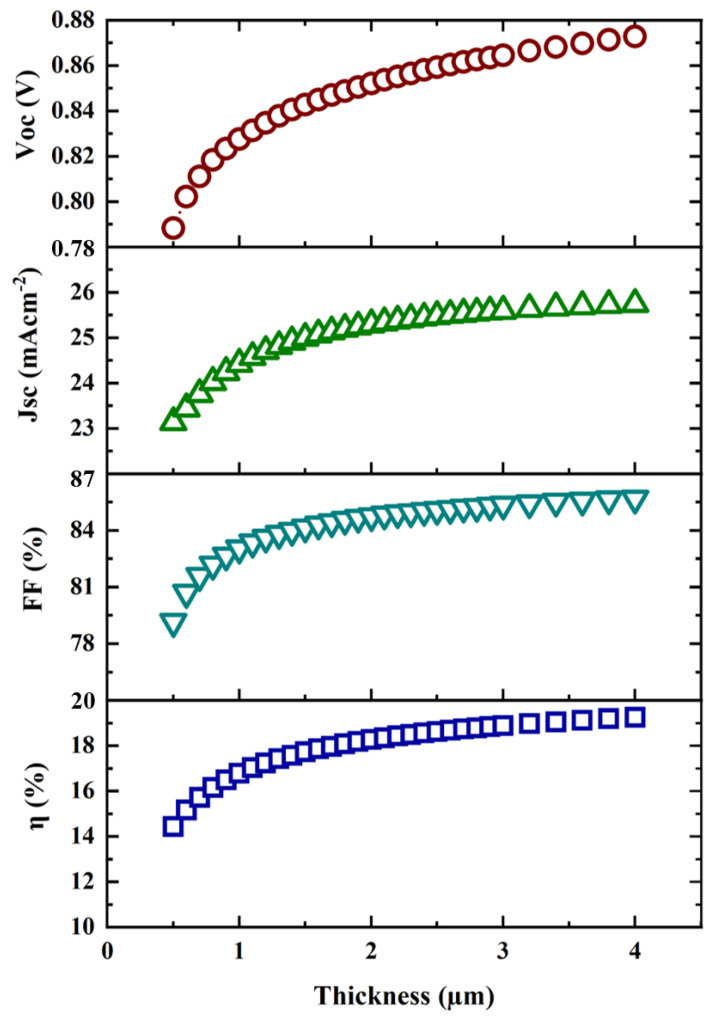
Absorber layer thickness versus V_oc_, J_sc_, FF, and η.

**Figure 5 micromachines-13-01249-f005:**
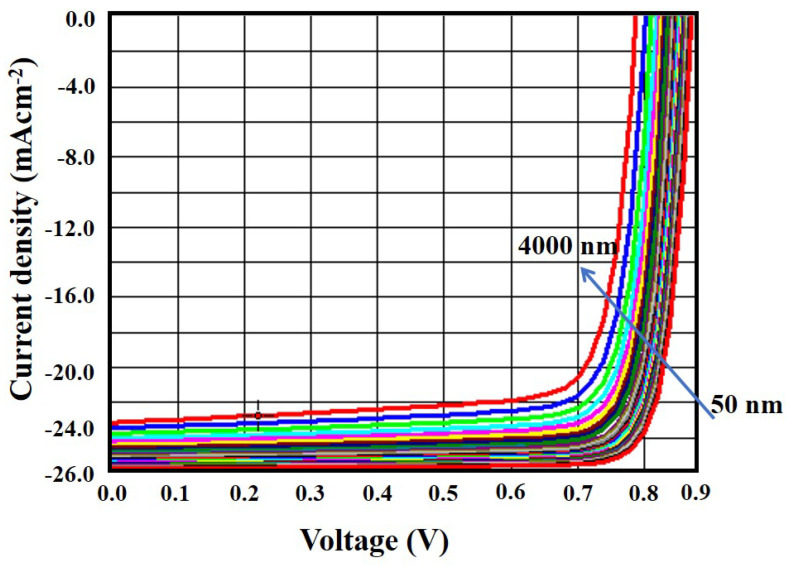
J–V curve with the variation of absorber layer thickness.

**Figure 6 micromachines-13-01249-f006:**
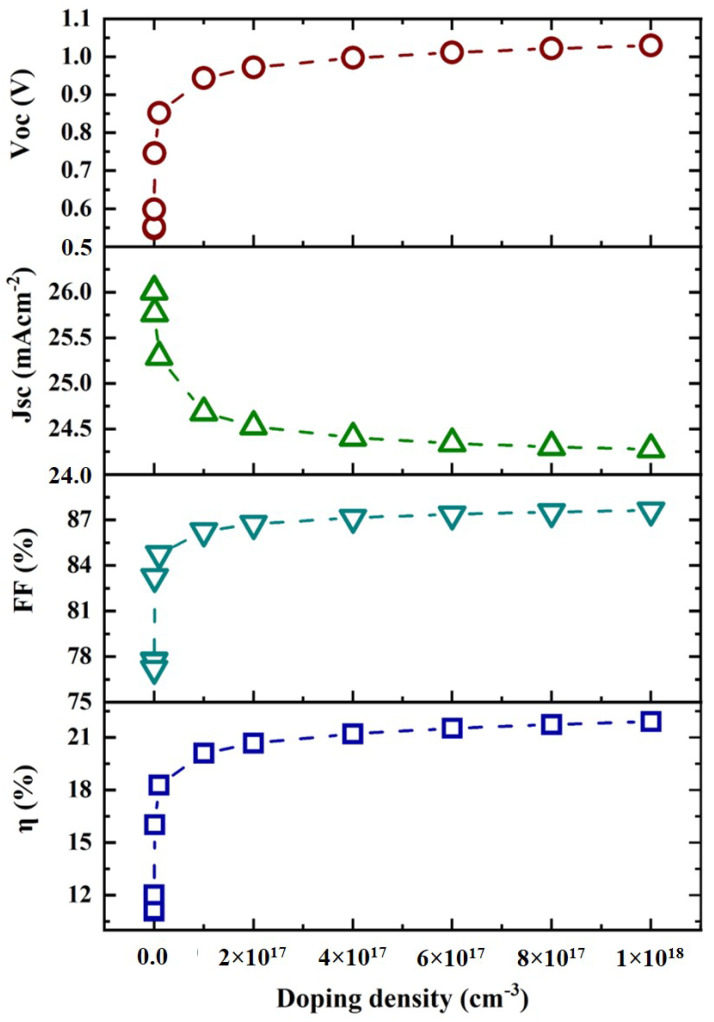
Doping density in the absorber layer versus V_oc_, J_sc_, FF, and η.

**Figure 7 micromachines-13-01249-f007:**
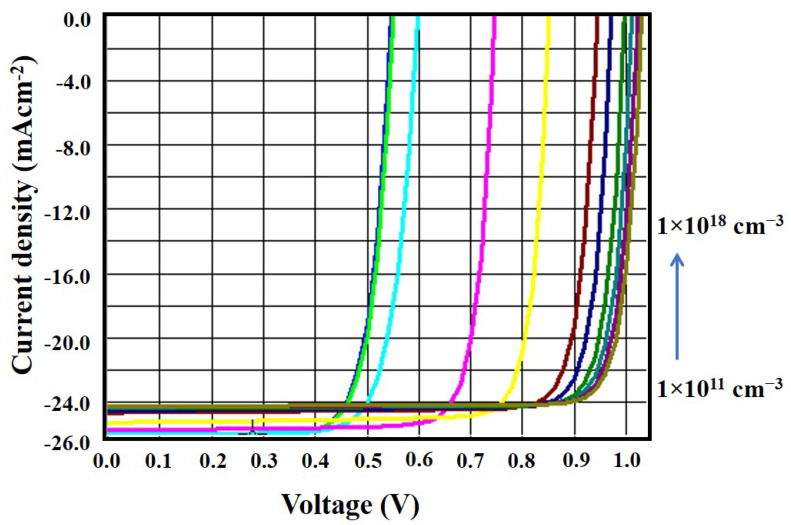
J–V curve with the variation of doping density in the absorber layer.

**Figure 8 micromachines-13-01249-f008:**
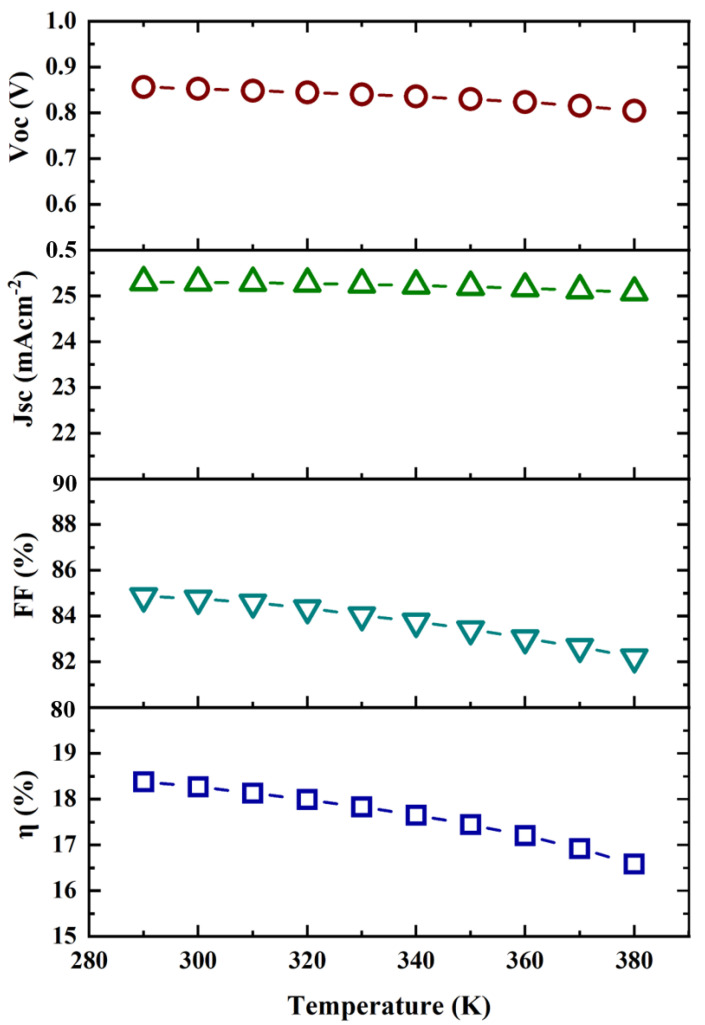
Effect of temperature versus V_oc_, J_sc_, FF, and η.

**Figure 9 micromachines-13-01249-f009:**
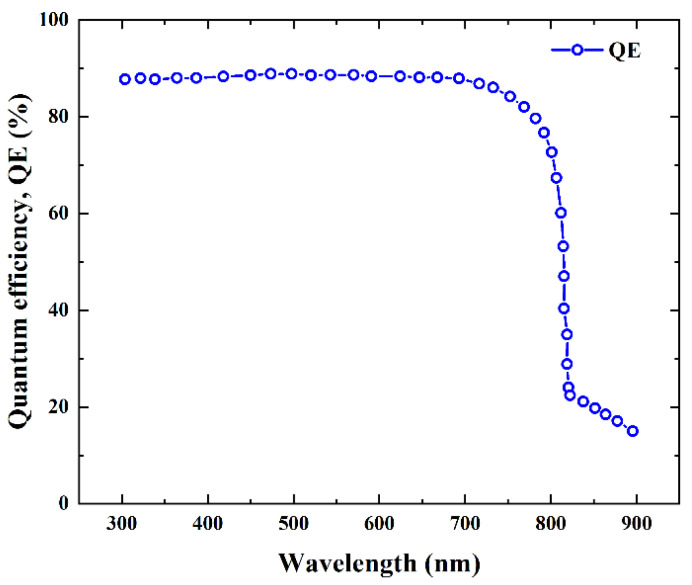
Quantum efficiency versus wavelength.

**Figure 10 micromachines-13-01249-f010:**
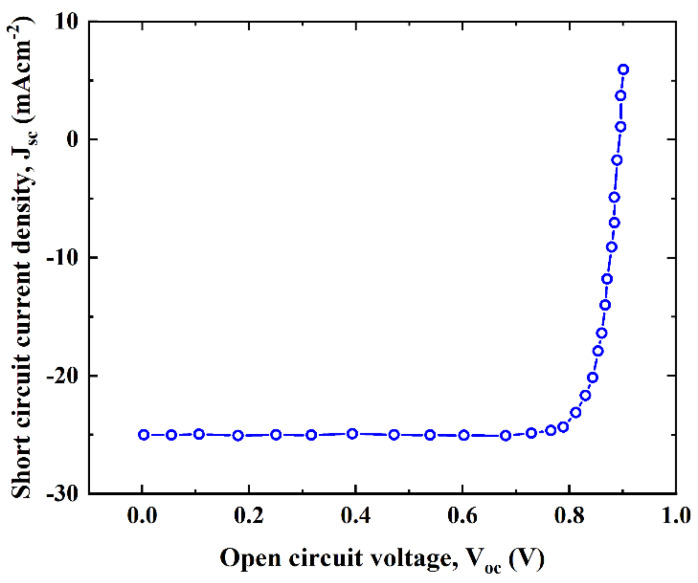
The J–V characteristics of a CZTS solar cell.

**Table 1 micromachines-13-01249-t001:** Input optical and electrical parameters of CZTS solar cell.

Parameters	Graphene	MoS_2_	CZTS
Energy bandgap (eV)	1.80	1.3	1.52
Electron affinity (eV)	3.92	4	4.50
Relative permittivity ${ɛ}_{r}$	10	4	10
Conduction band effective density of states (×10^18^ cm^−3^)	1000	0.75	2.2
Valance band effective density of states (×10^19^ cm^−3^)	100	0.18	1.8
Electron thermal velocity (×10^7^ cm s^−1^)	5.2	0.01	1.0
Hole thermal velocity (×10^7^ cm s^−1^)	5.0	1.0	1.0
Electron mobility (cm^2^ v^−1^ s^−1^)	1.0 × 10^9^	100	100
Hole mobility (cm^2^ v^−1^ s^−1^)	10	150	20
Doping density (×10^18^ cm^−3^)	9000	1000	0.01

**Table 2 micromachines-13-01249-t002:** Values were obtained from numerical analysis of the open-circuit voltage (V_oc_), short-circuit current density (J_sc_), fill factor (FF), and power conversion efficiency (η).

Solar Cell Structure	V_oc_ (V)	J_sc_ (mA cm^−2^)	FF (%)	η (%)	Ref.
Graphene/MoS_2_/CZTS	0.85	25.30	84.76	18.27	This work
Graphene/ZnO/CZTS	0.85	23.81	84.70	17.14	[16]
i-ZnO/MoS_2_/CZTS	1.01	29.42	57.40	17.03	[14]
ZnO/InSe/CZTS	1.00	28.06	58.40	16.30	[7]
Al:ZnO/i-ZnO/CdS/CZTS	0.78	27.98	72.86	15.84	[12]
ZnO/ZnS/CZTS	0.64	23.96	65.20	10.00	[41]
ZnO/CdS/CZTS	0.61	26.95	57.20	9.47	[41]

## Data Availability

Not applicable.

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
