# Peer review of "Numerical Analysis of a CZTS Solar Cell with MoS2 as a Buffer Layer and Graphene as a Transparent Conducting Oxide Layer for Enhanced Cell Performance"

_micromachines, 2022, doi:10.3390/mi13081249_

Round 1

Reviewer 1 Report

This paper is about CZTS solar cells. Calculations using the SCAPS show the performance of solar cells. This software is popular, but it is also known to be inconsistent with experimental data. The author is changing the film thickness for new CZTS solar cells using future materials. It is very important and it is understandable that there is no literature to compare them. In addition, the author has previously reported several material and structural treatises on CZTS. This is an extension of the treatise, but it is not easy to understand. At the very least, there should be a clear distinction between experimental and computational data. In particular, the contents of the introduction and Table 2.

Author Response

Thank you for reviewing our manuscript. Please find below a detailed response to the comments.

Reviewer 2 Report

The present work performs a numerical analysis of CZTS solar cells with MoS2 as a buffer layer and graphene as a transparent conductive oxide layer. Overall the work is well written and well designed. However, some improvements are needed. Below I leave my suggestions for improvements.

i) In the introduction, it is necessary to cite the references: https://doi.org/10.3390/cryst11121468 and https://doi.org/10.1016/B978-0-12-821592-0.00020-0.

ii) It is very important to add J-V curves for 1) Effect of Buffer Layer Thickness; 2) Effect of Absorber Layer Thickness, and 3) Effect of Doping Density of CZTS Absorber Layer.

After these improvements, the work can be published.

Author Response

(The authors gave the same response as above.)

Reviewer 3 Report

1. NREL has reported the CZTS solar cell a record PCE of 13%. The record can't be improved for several years. This work focus on simulating the Graphene/MoS2/CZTS solar cell with a PCE of 18.27%. I hope author can provide more experimental results to support this simulation results.

2. Fig.4. No data points present in the range of 1E17 and 1E18 of doping density. It is better to show more points in this range.

3. I suggest to discuss more math models to help readers understand how these input parameters can obtain simulation results.

Author Response

(The authors gave the same response as above.)

Round 2

Reviewer 2 Report

The authors made all the suggested modifications. Therefore, the article can be accepted for publication in its present form.

Reviewer 3 Report

The revised version is acceptable.